# Endometrial Status in Queens Evaluated by Histopathology Findings and Two Cytological Techniques: Low-Volume Uterine Lavage and Uterine Swabbing

**DOI:** 10.3390/ani11010088

**Published:** 2021-01-05

**Authors:** Alba Martí, Anna Serrano, Josep Pastor, Teresa Rigau, Ugné Petkevičiuté, Maria Àngels Calvo, Esteban Leonardo Arosemena, Aida Yuste, David Prandi, Adrià Aguilar, Maria Montserrat Rivera del Alamo

**Affiliations:** 1Department of Animal Medicine and Surgery, Veterinary School, Universitat Autònoma de Barcelona, 08193 Bellaterra, Spain; g21alba@gmail.com (A.M.); anna95serrano@gmail.com (A.S.); teresa.rigau@uab.cat (T.R.); petkeviciute.ugne23@gmail.com (U.P.); david.prandi@uab.cat (D.P.); adria.aguilar@uab.cat (A.A.); mariamontserrat.rivera@uab.cat (M.M.R.d.A.); 2Department of Anatomy, Veterinary School, Universitat Autònoma de Barcelona, 08193 Bellaterra, Spain; mariangels.calvo@uab.cat (M.À.C.); estebanleonardo.arosamena@uab.cat (E.L.A.); aidayuste@gmail.com (A.Y.)

**Keywords:** uterus, cytology, polymorphonuclear neutrophil, queen

## Abstract

**Simple Summary:**

The endometrium health of feline queens can be difficult to assess due to the reduced size of the uterus, which hinders representative biopsy sampling. This may result in limitations in diagnosing endometritis, and consequently in detecting infertility problems. Although histology is considered the most reliable technique for diagnosing endometritis in many species, cytology is also gaining importance and may be an alternative tool for evaluating the endometrium in small species. Two different common cytological techniques (uterine lavage and uterine swabbing) were compared to determine the reliability of cytology for evaluating the endometrium status in queens. Histopathological and bacteriological information was used for the control methods. Our results demonstrated that cytology may be a useful diagnostic tool for assessing the endometrial status. In addition, when comparing cytological techniques, the uterine lavage method was more representative than uterine swabbing.

**Abstract:**

Endometritis is associated with fertility problems in many species, with endometrial biopsy being the main diagnostic tool. In feline queens, the reduced size of the uterus may make it difficult to obtain representative diagnostic samples. Endometrial cytology may represent a valuable diagnostic tool for evaluating the health status of the endometrium in queens. Fifty domestic shorthair queens were included and divided into two cytological diagnostic technique groups, the uterine lavage (UL; n = 28) and uterine swabbing (US; n = 22) groups. Cytological results were compared with histopathological and bacteriological information. Changes in the histopathological patterns were also evaluated and compared with progesterone levels to confirm previous published data. Furthermore, the results from both cytological sampling methods were compared to evaluate the utility of each method. Endometritis was ruled out in all queens by means of histology and microbiology. Leukocyte counts and red blood cell/endometrial cell ratios were significantly higher in US than UL samples. Additionally, UL sampling is less affected by blood contamination and cells are better preserved. The combination of endometrial cytology and uterine culture might be useful for evaluating the endometrial characteristics in queens. The UL evaluation method is more representative of the actual endometrial status than the US technique.

## 1. Introduction

Endometritis is defined as inflammation of the uterine mucosa not extending deeper than the stratum spongiosum [1]. It has been widely associated with infertility, recurrent implantation failure, impaired embryo survival, and pregnancy loss in many species, such as mares [2,3], cows [4,5], bitches [6], and women [7].

However, endometritis is not always associated with evident clinical signs, with infertility being the only detected alteration [6,8,9]. In fact, recent publications on bitches support the notion that subclinical endometritis is a frequent finding in clinically healthy animals [6]. In cows, this condition is well defined and animals that undergo endometritis without evidence of any other clinical sign are also recognized as having subclinical endometritis, also named cytological endometritis [5,10]. In subclinical endometritis, endometrial cytology is characterized by the presence of polymorphonuclear neutrophils (PMN) with absent or minimal intrauterine exudate [5,11,12]. In human medicine, asymptomatic endometritis is known as chronic endometritis, and it is also associated with recognized fertility problems. It has a specific histological pattern that is characterized by the presence of a subtle lymphoplasmacytic inflammation of the endometrium [7]. In queens, studies on endometritis are scarce and previous authors agree that subclinical endometritis is difficult to evaluate [13,14], hampering elucidation of the actual prevalence in this species. The exact etiology of subclinical and chronic endometritis in healthy females has not been completely understood and seems to be different depending on the species [6].

In most species, endometritis is diagnosed by means of cytological and bacteriological evaluation of endometrial samples, in combination with ultrasonographic and histopathological assessment [15,16,17], being the latest and most reliable tools used to diagnose endometritis in many species, such as mares [18,19], bitches [20], and cows [21].

In queens, the reduced size of the uterus may make it difficult to obtain representative tissue samples for histopathological examination. This may limit the diagnosis of possible endometritis. Thus, finding an easy, safe, and reliable technique other than endometrial biopsy to evaluate the endometrium in queens would open the possibility for the diagnosis of endometritis in this species. Furthermore, endometrial cytology is gaining importance as a diagnostic technique. In cows, some authors consider endometrial cytology as a potential reference test because it can detect both clinical and subclinical endometritis [11]. In other species such as mares [9] and women [22], endometrial cytology is also considered a valuable tool for the diagnosis of certain uterine diseases. No information exists in cats using this diagnostic technique.

The main aim of this study was to determine which cytological sampling method, either (1) low-volume uterine lavage (UL) or (2) uterine swabbing (US), is more reliable for evaluating endometrial status in queens. Histological and microbiological analyses were used as control techniques to confirm the actual endometrial status.

## 2. Materials and Methods

### 2.1. Animals

A total of 50 domestic shorthair queens of different ages were included in this study. Queens belonged to a neutering program of stray cats carried out by the Surgery Unit of the Facultat de Veterinària of the Universitat Autònoma de Barcelona (UAB). No information about the previous reproductive history, age, or reproductive status of the queens was available. Before the inclusion, a complete physical examination and tests against feline leukemia virus and feline immunodeficiency virus (Snap FIV/FeLV; Idexx Laboratories, Westbrook, ME, USA) were performed. Only those animals that were healthy and negative against both diseases were included in the study.

Prior to the surgery, queens were pre-medicated with an intramuscular combination of 5 mg/kg ketamine (Ketamidor®, Richer pharma) with 20 μg/kg buprenorphine (Buprecare®, Divisa FramaVic, S.A) and 0.2 mg/kg midazolam (Midazolam®, Normon S.A). When each animal reached sedation status, a blood sample from the jugular vein was collected and placed into a glass tube. The blood was allowed to clot and then centrifuged at 2200× *g* for 10 min at room temperature. The serum was immediately frozen and kept at −20 °C until analyses were performed.

After blood collection, queens were induced intravenously with 2–4 mg/kg propofol (Propovet TM, Zoetis Ecuphar). Anesthetic status was maintained using 1.5–2% isoflurane (IsoFlo, Zoetis Ecuphar) in oxygen using a Mapleson F anesthetic breathing circuit. When the anesthetic status was reached, a conventional ovariohysterectomy was performed via ventral midline laparotomy.

After ovariohysterectomy was performed, uteri were randomly divided into two groups according to the technique used to obtain the cytological preparations, namely the uterine lavage group and uterine swabbing group. The uterine lavage group (UL) included a total of 28 queens, while the uterine swabbing group (US) included a total of 22 queens.

The experimental procedure was approved by the Ethical Committee for Animal Care and Research of the UAB (CEEAH, code 2939).

### 2.2. Sample Collection

In the UL group, a total of 2 mL of phosphate-buffered saline (PBS) was injected into the uterine lumen through the uterine wall using a 2 mL sterile syringe with a 23G needle. The solution was instilled and at least 0.5 mL was aspirated back. The first drop was discarded and the next 2–3 drops were used for microbiological purposes. A sterile swab (Eurotubo®, invasive sterile collection swab, Deltalab; Rubí, Spain) was soaked and then introduced into a transport medium tube. The remaining volume was then placed in a 2 mL Eppendorf tube for the cytological evaluation, which was processed within 1 h of collection. A 50 μL aliquot was centrifuged at 3330× *g* for 5 min (Rotofix 32A, Andreas Hettich GmbH and Co, Tuttlingen, Germany) and a smear from the pellet was performed.

In the US group, a longitudinal incision with a scalpel blade (Swann-Morton®, sterile carbon steel surgical blades; Sheffield, England) in the uterine horn was performed to gain access to the uterine lumen. A sterile swab (Eurotubo®, invasive sterile collection swab, Deltalab; Rubí, Spain) was then introduced through the incision and rotated to collect a sample from the endometrium surface. To preserve the sample, the swab was introduced into a transport medium tube and submitted for microbiological exam. After this, the incision was longitudinally elongated and another sterile swab was introduced and rotated to collect another sample. The sample was immediately smeared on a sterile glass slide to obtain the cytological sample.

Once samples for cytology and microbiology were obtained, a full-thickness biopsy for histopathological study was obtained and immediately placed in 10% paraformaldehyde.

### 2.3. Progesterone

Serum progesterone (P_4_) was measured in order to establish the ovulation status of the queens. Those queens with progesterone concentrations below 1.5 ng/mL were considered non-ovulated, while those showing progesterone levels above 1.5 ng/mL were considered ovulated [23,24,25,26]. Serum progesterone concentration was determined using an Inmulite 1000 instrument (Immulite; Siemens Healthcare Diagnostics, Cornellà del Llobregat, Barcelona, Spain). The intra-assay and inter-assay coefficients of variation (CV %) were 5.5 and 6.5, respectively.

### 2.4. Microbiology

Microbiological evaluation was performed following the protocols established by the Applied and Environmental Microbiology Laboratory of Animal Health and Anatomy Department of the Veterinary Faculty (UAB). Samples were grown in blood agar (Columbia Agar supplemented with sheep blood), Man–Rogosa–Sharpe agar (MRS, tryptic soy agar (TSA), McConkey agar (MK), Baird–Parker agar (BP) supplemented with egg yolk tellurite emulsion, Sabouraud agar supplemented with 0.5 gr/L chloramphenicol, and tryptone sulfite neomycin (TSN) agar. All culture media were purchased from Liofilchem Srl (Italy), with the exception of Columbia agar, which was purchased from Bio-Rad Laboratories (USA). Oxygen and temperature conditions for the different culture media are shown in Table 1.

Samples were processed within one hour of sampling. Swabs were streaked on Petri dishes following a continuous streak method. Then, plates were incubated for 24 h at the different stated conditions and an initial counting of colonies was performed. Those plates with negative growth were incubated during 24 more hours and the counting was repeated at the end of this second period of incubation. Plates in anaerobic conditions were incubated for 48 h with an Anaerocult system (Merck KGaA, 64271 Darmstadt, Germany), whereas Sabouraud agar plates were incubated for 7 days before being discarded.

### 2.5. Histopathology

Here, 4-μm-thick sections were obtained for histological examination. Samples were then stained with hematoxylin and eosin and evaluated according to previously reported data [27] through the NPD View 2 viewing software (Hamamatsu Photonics, Korea). The biopsy assessment included evaluation of the glandular density and diameter, height of the luminal epithelia, and white blood cell counts.

Glandular density was evaluated by counting the number of glands in 10 randomly selected fields at 400× magnification in both the stratum spongiosum and the stratum compactum. Glandular diameter was evaluated by considering the mean of two perpendicular diameters of each gland (from the basal lamina to the opposite one), measured in the stratum spongiosum. A total of 30 randomly selected glands were studied at 400× magnification. The height of the luminal epithelia was determined by measuring a total of 30 randomly selected cells from the basal lamina to the apical membrane, recorded at 400× magnification. Neutrophil counting was performed by counting in 10 randomly selected fields at 400× magnification. A mean for each variable was calculated for each sample.

### 2.6. Cytology Samples

Both UL and US samples were stained using an automatic dye (Hematek®, Siemens Healthcare Diagnostics INC, Tarrytown, NY, USA) with a modified Giemsa stain (Auto-Hemacolor®, Merck KGaAQ, Darmstadt, Germany), then examined using a light microscope at ×100 to ×1000 magnification (Nikon Eclipse Ci, Nikon, Japan). Slides were examined separately for cytological analysis by a board-certified pathologist and a resident of clinical pathology. Endometrial cells, erythrocytes, and leukocytes were examined. The analyzed characteristics and classification scores are summarized in Table 2. The parameters studied from each sample were the smear quality, the smear cellularity (including the characterization of the endometrial cells), the calculated cells scores, and the evaluation of inflammation focusing on the presence and disposition of polymorphonuclear neutrophils (PMN), as described below. For each smear, the mean of both observed results was used. When a discrepancy existed, a consensual result was obtained.

#### 2.6.1. Smear Quality

The smear quality was assessed by evaluating the background, the blood dilution of the sample, and the presence of endometrial cells. The background was graded with a 0 to 3 score according to the presence of mucus or proteinaceous debris. A 0 score was given to smears without mucus or debris, a score of 1 was applied for those with the presence of mucus or proteinaceous debris that were adequate for diagnosis, a score of 2 was given to smears with abundant mucus or proteinaceous debris that were still adequate for diagnosis, and finally a score of 3 was given to samples with excessive presence of mucus and proteinaceous debris that were not adequate for diagnosis.

The erythrocyte content or blood dilution of the sample was evaluated using a similar approach as previously described [28]. A numerical score ranging from 0 to 4 was applied according to the number and distribution of red blood cells. A score of 0 was obtained when none or very few erythrocytes (0–5 RBCs/40× magnification) were present in the preparation. Smears scored as 1 were those with few dispersed erythrocytes (6–50 RBCs/40×). Samples scored as 2 showed moderate numbers of red blood cells, either dispersed or in small clumps (51–150 RBCs/40×). Those with a score of 3 showed increased numbers of erythrocytes frequently organized in different sized clumps (151–300 RBCs/40×). Finally, smears scored as 4 showed abundant erythrocytes mostly in big clumps (>300 RBCs/40×).

The examination of the smear quality was also performed based on a rapid evaluation of the presence of endometrial cells, including those cells that exfoliate individually and those that were in clusters. These was generally categorized as the presence or absence of endometrial cells. The quantity of endometrial cells was evaluated in the cellularity category. Regarding endometrial cells, preservation was also graded by determining the presence of broken cells. A score of 0 or adequate was assigned when no broken cells were present, a score of 1 or good preservation was assigned when there were less than 25% broken cells, a score of 2 or fairly adequate was assigned when there were 25–50% broken cells, and a score of 4 or not useful for diagnosis was assigned when more than 50% of the endometrial cells were broken.

#### 2.6.2. Cellularity and Cell Morphology

Epithelial endometrial cells were examined and scored from 0 to 4 (0 = no cells; 1 = very few cells; 2 = few cells; 3 = moderate number of cells; 4 = abundant cells), as previously described [28]. Cell morphology was determined by the observation of different features, including the degree of preservation (presence of pyknotic cells), disposition (individual, clusters, or acini formation), morphology (round, low-columnar, columnar), presence of cytoplasmic vacuoles or cilia, presence of a prominent nucleolus, and number of mitoses similar to that previously described in the bitch specimen [29].

#### 2.6.3. Cell Ratios

Cell scores were used for the evaluation of the erythrocyte and leukocyte contents of the samples.

The presence of erythrocytes (RBC) was reported and ranged as described above in terms of the smear quality. Furthermore, an erythrocyte/endometrial cell ratio was calculated according to previously reported data [28] with slight modifications. Briefly, the ratio was assessed by determining the mean number of epithelial cells obtained by counting the number of epithelial cell clumps in 10 power fields at 400× magnification. Then, the mean was divided to obtain the erythrocyte score.

The presence of leukocytes was also evaluated and ranged from 0 to 4 (0 = no leukocytes; 1 = very few leukocytes; 2 = few leukocytes; 3 moderate number of leukocytes; 4 = abundant leukocytes) as previously described [28] with some modifications. A differential white blood cell count was also made from at least 100 leukocytes when possible.

#### 2.6.4. Inflammation

The presence or absence of inflammation was made on basis of the PMN ratio according to previously defined criteria [30]. Briefly, 10 fields at 40× magnification were visualized and samples were scored in 4 categories: category 0 (no PMN); category +1 (1 to 10 PMNs individually distributed); category 2+ (more than 10 PMNs individually distributed); category 3+ (large clumps of PMNs). Preparations in category 0 were considered as not having inflammation, those in category 1 were considered to have mild inflammation, those in category 2 were considered to have moderate inflammation, and finally those in category 3 were considered to have severe inflammation.

### 2.7. Statistical Analysis

Variables were evaluated using the statistical package SPSS (version 22, IBM Corporation, Armonk, NY, USA). A Shaphiro–Wilk test was used to study the normal distribution of the variables. For those parameters with a non-parametric distribution, a Wilcoxson signed-rank test was applied to compare two related samples. Independent measurements were compared using the Mann–Whitney U test. For those parameters with a normal distribution, a two-way ANOVA test was used.

## 3. Results

### 3.1. Progesterone

Serum levels of progesterone ranged from 0.3 to 19.7 ng/mL. In UL queens the values ranged from 0.3 to 19.7 ng/mL, while in US queens they ranged from 0.8 to 3.0 ng/mL. Non-ovulated queens showed a mean value of 0.9 ± 0.1 ng/mL (ranging from 0.3 to 1.2 ng/mL) in the UL group and of 1.0 ± 0.1 ng/mL (ranging from 0.8 to 1.3 ng/mL) in the US group. Regarding the ovulated queens, mean serum progesterone values of 10.6 ± 1.3 ng/mL (ranging from 1.9 to 19.7 ng/mL) and 2.3 ± 0.5 ng/mL (ranging from 1.5 to 3.0 ng/mL) were obtained for the UL and US groups, respectively (Table 3).

### 3.2. Microbiology

All samples yielded negative growth.

### 3.3. Histology

Histopathologic study of all samples did not identify any queens with signs of endometritis. Regarding the histological parameters studied, the results showed that the epithelial height in animals with high levels of progesterone (*p*_4_ > 1.5 ng/mL) was significantly (*p* < 0.05) higher than those of animals with low levels of serum progesterone (*p*_4_ < 1.5 ng/mL) (Table 4 and Figure 1).

There was no significant difference in the glandular density (*p* = 0.278) between groups (Table 4 and Figure 2). However, the glandular diameter was significantly (*p* = 0.012) increased in the high progesterone group compared with the low progesterone group (Table 4 and Figure 3).

Finally, the neutrophil counts were low in both groups and there were no statistically significant differences related to progesterone levels, although queens with high levels of progesterone showed a trend for lower neutrophil counts (Table 4 and Figure 4).

### 3.4. Cytology

When evaluating the smear quality, cellularity, and morphology, although no statistically significant differences were observed between sampling methods, the preservation and cell morphology was detected to be moderately better in the samples obtained with the lavage method (Table 5 and Table 6).

Regarding cell scores, only the RBC/endometrial cell ratio and white blood cells (WBC) count were significantly different between methods (Figure 5, Figure 6 and Figure 7), being higher in samples obtained by uterine swabbing than those obtained by uterine lavage.

Furthermore, the numbers of PMN cells were not statistically significant between the evaluated methods (Table 7), although increased percentages of swabbing samples were classified as category 2+. In swabbing samples, 9% of the samples were classified as category 0, 36% as category 1, 50% as category 2, and 5% as category 3. In lavage samples, 32% were classified as category 0, 39% as category 1, 29% as category 2, and no samples were classified as category 3 (Table 7).

Finally, regarding inflammation, none of the samples showed increased numbers of neutrophils so as to suggest endometritis, regardless of the sampling group.

When different levels of progesterone were considered (i.e., ovulated versus non-ovulated queens), no significant differences in the cytological analysis were observed between sampling groups.

## 4. Discussion

Endometritis diagnosis is performed by combining different diagnostic tools such as biopsy, cytology, and culture, among others, with histology classically being used as a fundamental tool for several mammalian species [18,19,20,21]. Histopathological findings have been used to categorize the severity of endometritis in bitches [20]. Animals with neutrophilic or eosinophilic infiltration are classified as having acute endometritis, those with mixed infiltration are classified as having subacute endometritis, while animals that have lymphocytic or lymphoplasmocytic infiltration are classified as having chronic endometritis. However, literature related to the study of endometrium and endometritis in queens is scarce.

Knowledge of normal uterine characteristics in a healthy subjects is mandatory to detect alterations that can contribute to fertility problems. In the present study, microbiological cultures and biopsies were used as controls to establish the actual status of the endometrium. All microbiological cultures yielded negative results and PMN counts were low in all of the samples, which allowed all of the analyzed endometria to be categorized as normal.

In addition, the endometrial structure was evaluated in the present study and modifications related to serum levels of progesterone were described. The epithelium and stroma undergo cyclic morphological and biochemical changes during the reproductive cycle, which are necessary to create the optimal environment for the survival of the fetus [31,32,33,34,35,36,37,38]. These changes are thought to be secondary to the actions and fluctuations in serum steroid hormones levels. In queens, three different histological patterns of the uterus (inactive phase, follicular phase, and luteal phase) according to the stage of the queen’s reproductive cycle have been previously described by Chatdarong et al. [39]. In that study, epithelial luminal height was significantly higher during the luteal phase (high progesterone and low estradiol serum levels) than during the inactive phase (low progesterone and estradiol serum levels), but not during the follicular phase (low progesterone and high estradiol serum levels). These results are in agreement with the present study, in which we observed increases of the epithelial height and the glandular diameter under the influence of high levels of serum progesterone. In our study, queens were classified according to serum progesterone levels, while estradiol levels were not considered, which could be a limitation, as we did not identify animals in the follicular phase or confirm than the epithelium heights did not differ from animals with high progesterone levels. Additionally, punctual determination of serum estradiol levels would not be representative of the reproductive status of a queen. Serialized blood sampling for hormone determination would be a more accurate way to determine the role of steroid hormones in the endometrium of a queen throughout the sexual cycle. Since the queens included in the study belonged to a neutering program, serialized blood sampling was beyond the authorized permission.

To the authors’ knowledge, this is the first study wherein histology includes the evaluation of the glandular density and glandular diameter in order to further characterize the histologic appearance of queens in different phases of the reproductive cycle. Our results agree with those previously observed in mares on day 7 of pregnancy [40]. Specifically, increased glandular diameter, height of the glandular epithelium, and glandular lumen were observed in 7-day-pregnant mares in comparison with cyclic mares. All of these changes have been observed to appear in concomitance with the entry of the embryo into the uterus. Focusing on the increased height of the luminal epithelium and glandular diameter, these changes have been suggested to be related with the production of glandular secretions, which are released into the uterine lumen during early pregnancy.

On the other hand, cytological examination of the endometrium is considered a useful technique and has gained importance as a part of the diagnosis of endometritis in different species, such as bitches, mares, and cows [1,10,41]. In the literature, different techniques have been described for the collection of endometrial and inflammatory cells. Uterine lavage and swabbing techniques are the most accepted cytological methods in cows, since they are less invasive [42]. Although histology is still considered the most reliable tool for diagnosing endometritis in many species such as mares [18,19], bitches [20] or cows [21], the reduced endometrium size of queens makes it difficult to obtain representative samples. Having a minimally invasive cytology technique that yields preserved endometrial and inflammatory cells of the endometrium surface would provide a more feasible tool for endometritis diagnosis.

In the present study, samples obtained with UL and US methods were evaluated to determine their applicability in order to obtain cellular samples from the feline endometrium. Furthermore, both techniques were compared to evaluate which one was more appropriate for establishing the actual status of the feline endometrium. Both sampling techniques collected enough endometrial cells to consider them as diagnostic. However, swabbing samples showed statistically significant higher RBC/endometrial cell ratios and higher WBC counts compared with lavage samples, with lavage samples closer to histological results than swabbing samples. These differences are probably due to the friction of the endometrial surface with the swab during the sampling, causing the rupture of superficial vessels and facilitating bleeding. The bleeding probably contributed to the increase of WBC percentages, as well as the higher RBC/endometrial cell ratios obtained. The present results demonstrate that endometrial smears obtained by uterine lavage are more reliable for establishing the status, at least when healthy, of the feline endometrium.

Although no significant statistical differences were observed, when evaluating the smear quality, cellularity, and morphology, cell preservation and morphology results were detected to be moderately better in the samples obtained with the lavage method. Furthermore, samples obtained with the US method showed higher scores in the PMN evaluation, although the differences were not sufficient to be statistically significant.

As previously stated, the aim of the present study was to determine which one of the sampling techniques (US vs. UL) was more accurate. It is also important to keep in mind the fact that while uterine lavage is a clinically applicable technique in queens [43], uterine swabbing is not.

In contrast with observations of histology, cytology results from the present study did not show any variation in the cell exfoliation degree or in the cell morphology according to serum progesterone levels.

The study has some limitations. First of all, queens included in this study were feral and information on their previous reproductive history, age, and reproductive status was not available. Another limitation was that none of the queens included in the study had endometritis; thus, further research including queens with endometritis is warranted to establish the real utility of endometrial cytology in pathological conditions.

## 5. Conclusions

In conclusion, although further studies including queens with endometritis are warranted, the combination of endometrial cytology and uterine culture could be used to evaluate the endometrial status of queens. The uterine lavage method is less affected by blood contamination, cells are better preserved, and the samples are more representative of reality when compared with US samples. Although serum progesterone did not affect the cytology results, histologically the present results confirm previous reported data on the increase of endometrial epithelium height. In addition, the glandular diameter was also increased in queens with high levels of serum progesterone. Endometrial remodeling is probably necessary for the proper implantation and development of pregnancy.

## Figures and Tables

**Figure 1 animals-11-00088-f001:**
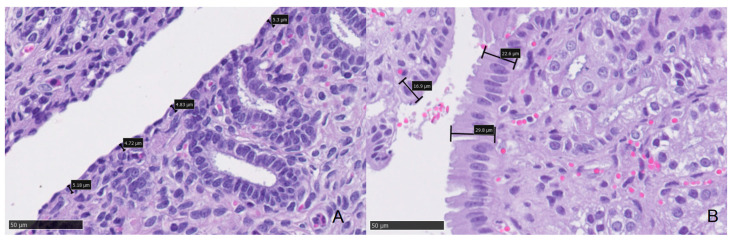
Epithelial height of a queen’s endometrium. (**A**) Endometrial biopsy from a queen with low levels of serum progesterone (1.4 ng/mL). Measurements obtained in the luminal epithelium of the endometrium ranged from 4.72 to 5.3 mm (black bars and boxes). (**B**) Endometrial biopsy from a queen with high levels of serum progesterone (7.7 ng/mL). Measurements obtained in the luminal epithelium of the endometrium ranged from 16.9 to 29.8 mm (black bars and boxes). Endometrial epithelial height was significantly higher in queens with high levels of serum progesterone.

**Figure 2 animals-11-00088-f002:**
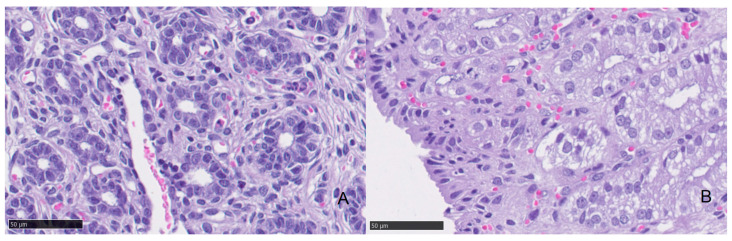
Glandular density in a queen’s endometrium. (**A**) Endometrial biopsy from a queen with low levels of serum progesterone (0.8 ng/mL). (**B**) Endometrial biopsy from a queen with high levels of serum progesterone (7.7 ng/mL). Glandular density scores showed no statistical differences between low and high progesterone groups.

**Figure 3 animals-11-00088-f003:**
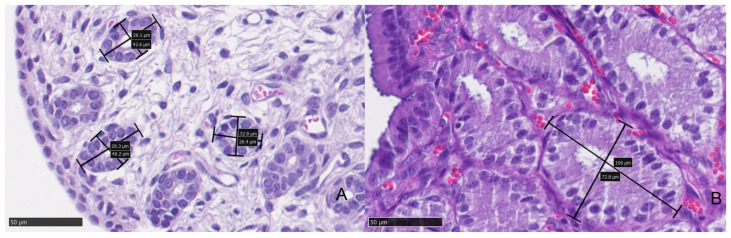
Glandular diameter in a queen’s endometrium. (**A**) Endometrial biopsy from a queen with low levels of serum progesterone (0.8 ng/mL). Black bars and boxes show the measures taken from the two perpendicular diameters of the gland (from the basal lamina to the opposite one), with a total mean of 33.5 mm in that queen. (**B**) Endometrial biopsy from a queen with high levels of serum progesterone (6.5 ng/mL). Black bars and boxes show the measures taken from the two perpendicular diameters of the gland (from the basal lamina to the opposite one), with a total mean of 98.99 mm in that animal. The glandular diameter values were significantly increased in animals with high levels of serum progesterone compared with the low progesterone group.

**Figure 4 animals-11-00088-f004:**
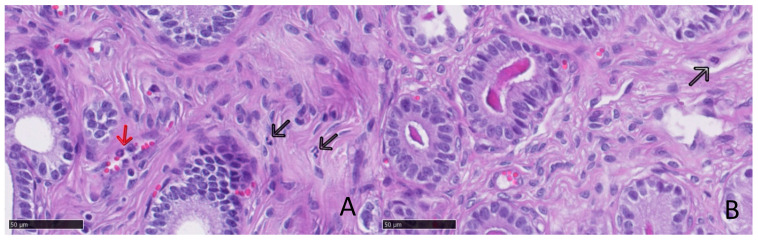
Presence of neutrophils in a queen’s endometrium. (**A**) Endometrial biopsy from a queen with low levels of serum progesterone (0.25 ng/mL). Note: Two neutrophils admixed within the endometrium (black arrow); erythrocytes and neutrophils within a blood vessel (red arrow). (**B**) Endometrial biopsy from a queen with high levels of serum progesterone (19.7 ng/mL). One neutrophil admixed within the endometrium can be observed (black arrow).

**Figure 5 animals-11-00088-f005:**
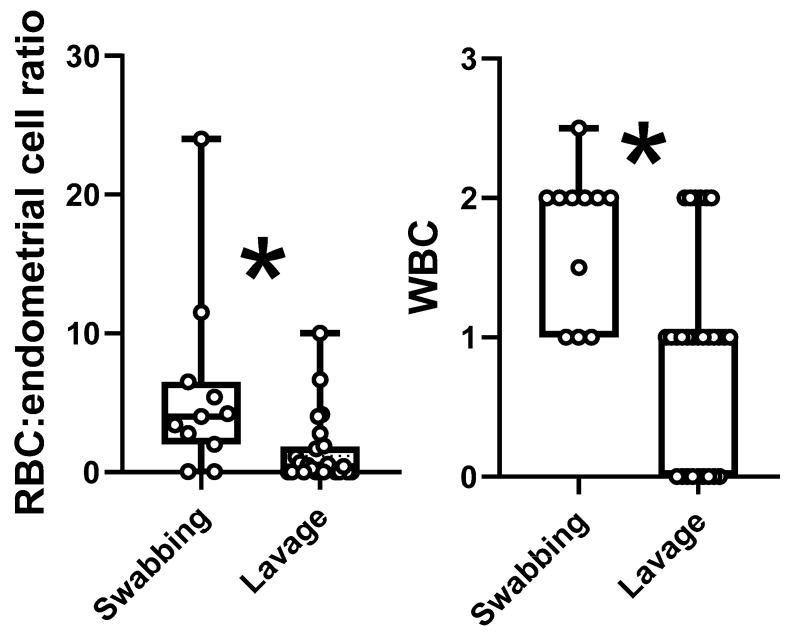
Red blood cells (RBCs): Endometrial cell ratio and white blood cell (WBC) numbers. * Samples obtained by uterine swabbing showed statistically significant (*p* < 0.05) higher values for these two analyzed parameters in comparison with samples obtained by uterine lavage.

**Figure 6 animals-11-00088-f006:**
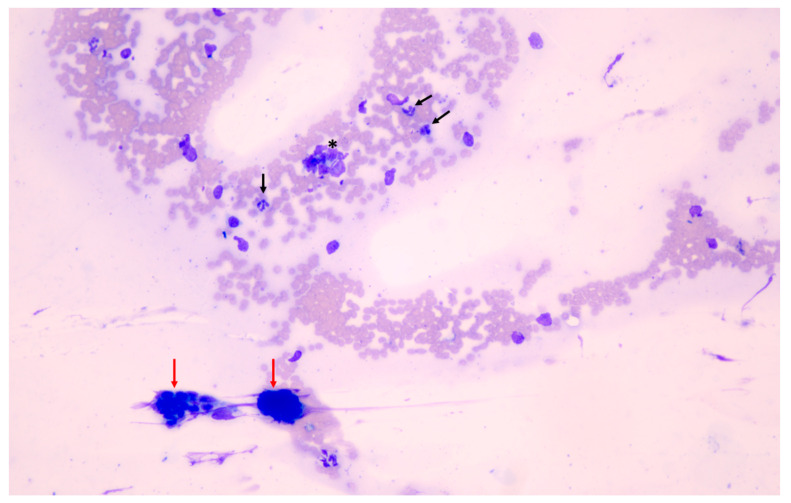
Endometrial cytology results obtained using the swabbing technique (200× magnification). Note the presence of abundant red blood cells (RBCs), which increases the RBC/endometrial cell ratio and the white blood cells content of the smear. Black arrows indicate the presence of neutrophils. Red arrows indicate clusters of endometrial cells. The asterisk indicates a broken cell.

**Figure 7 animals-11-00088-f007:**
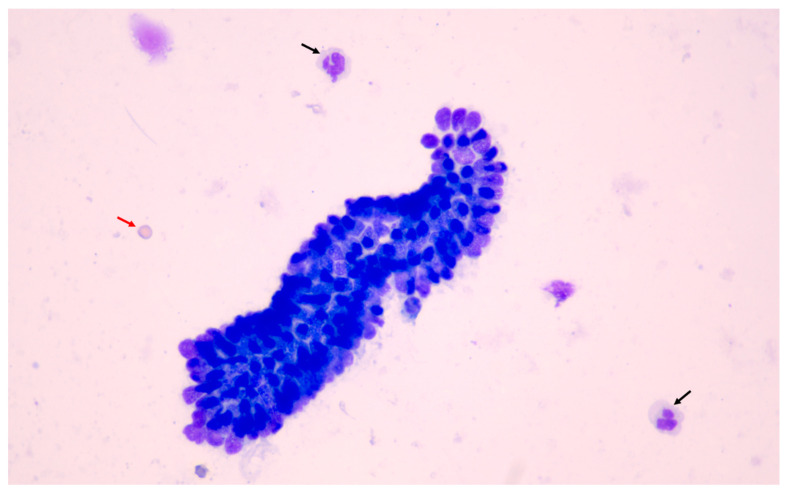
Endometrial cytology results obtained using the lavage technique (400× magnification). A big cluster of endometrial cells was detected and cell morphology was well preserved. Note the low numbers of red blood cells (red arrow) and leukocytes (black arrows).

**Table 1 animals-11-00088-t001:** Oxygen and temperature conditions for the different culture media.

Agar Media	Oxygen Conditions	Temperature
Blood	Aerobic	37 °C
	Anaerobic	37 °C
	5% CO_2_	37 °C
MRS	Aerobic	37 °C
	Anaerobic	37 °C
	5% CO_2_	37 °C
TSA	Aerobic	37 °C
McConkey	Aerobic	37 °C
Baird–Parker	Aerobic	37 °C
Sabouraud	Aerobic	28 °C
TSN	Anaerobic	42 °C

MRS: Man-Rogosa-Sharpe agar; TSA: tryptic soy agar; TSN: Sabouraud agar supplemented with 0.5 gr/L chloramphenicol and tryptone sulfite neomycin agar.

**Table 2 animals-11-00088-t002:** Cytologic features evaluated in the smears obtained with the uterine lavage and the uterine swabbing sampling methods.

Cytological Feature	Score				
	0	1	2	3	4
Mucus or Proteinaceous debris	Absent/clear background	Present but adequate for diagnosis	Abundant	Excessive, not adequate for diagnosis	-
Blood in Background (RBC/40×)	Absent (0–5)	Scattered (6–50)	Few (51–150)	Moderate (151–300)	Abundant (>300)
Cellularity	Absent	Scattered	Moderate	Abundant	-
Cell Preservation (% of broken cells)	Adequate (0)	Good (<25)	Fairly adequate (25–50)	Not useful for diagnosis (>50)	-
Epithelial Cells (number of cells)	Absent (0)	Scant (1–40)	Few (40–80)	Moderate (80–199)	Abundant (>200)
Individual Epithelial Cells	Absent	Scant	Few	Moderate	Abundant
Clusters Epithelial Cells (number of clusters)	Absent (0)	Scant (<10)	Few (10–40)	Moderate (40–80)	Abundant (>80)
Acinar Arrangement (HPF)	Absent (0)	Scattered <1/10 HPF	Moderate 1–3/10 HPF	Abundant >3/10 HPF	
Shape	Round	Low Columnar	Columnar		
Pyknotic (%)	Absent (0)	Scant (<10)	Few (10–20)	Moderate (20–50)	Abundant (>50)
Cilia	Absent (0)	Scattered <1/10 HPF	Moderate 1–3/HPF	Abundant >3/10 HPF	
Vacuoles (%)	Absent (0)	Scattered (<20)	Moderate (20–60) Cells	Abundant (>60) Cells	
Nucleoli (%)	Absent (0)	Scattered <20	Moderate 20–60	Abundant >60	
Mitosis (Number of mitoses)	Absent (0)	Scattered 0–1/10 HPF	Moderate 1–3/10 HPF	Abundant >3/10 HPF	

HPF: high power field. RBC: red blood cells.

**Table 3 animals-11-00088-t003:** Serum levels of progesterone in the different evaluated groups. Values are expressed as means ± SEM (range).

Serum Progesterone (ng/mL)	Uterine Lavage	Uterine Swabbing
Ovulated	10.6 ± 1.3 (1.9–20.0) (n = 15)	2.3 ± 0.5 (1.5–3.0) (n = 6)
Non-ovulated	0.9 ± 0.1 (0.3–1.2) (n = 13)	1.0 ± 0.1 (0.8–1.3) (n = 16)

**Table 4 animals-11-00088-t004:** Results for histological examination. Values are expressed as the median plus the minimum and maximum.

	Low Progesterone			High Progesterone			
	Median	Minimum	Maximum	Median	Minimum	Maximum	*p*
Epithelial Height (cm)	7.95	5.22	11.40	15.23	6.36	31.22	0.000
Diameter 1 (cm)	53.99	28.03	89.47	86.97	40.00	158.33	0.039
Diameter 2 (cm)	37.67	22.33	54.57	53.04	23.39	90.36	0.039
Mean Diameter (cm)	45.82	25.18	68.74	70.00	36.10	124.35	0.012
Glandular Density (glands/400×)	6.12	0.95	11.40	7.49	0.50	18.20	0.278
Neutrophil (in 10 fields 400×)	0.49	0.00	4.60	0.16	0.00	0.60	0.053

**Table 5 animals-11-00088-t005:** Grading and distribution of the smear quality parameters according to the uterine sampling method. No statistically significant differences were observed for any of the evaluated parameters.

Smear Quality	Technique	Score 0	Score 1	Score 2	Score 3	Score 4	Comments
Mucus or Proteinaceous debris	Lavage	28 (100%)	0 (0%)	0 (0%)	0 (0%)	-	
	Swabbing	4 (18.18%)	18 (81.81%)	0 (0%)	0 (0%)	-	
Blood in Background (RBC)	Lavage	9 (32.14%)	8 (28.57%)	4 (14.29%)	4 (14.29%)	3 (10.71%)	
	Swabbing	0 (0%)	6 (27.27%)	6 (27.27%)	10 (45.45%)	0 (0%)	
Cellularity	Lavage	6 (21.42%)	9 (32.14%)	8 (28.57%)	5 (17.85%)	-	
	Swabbing	0 (0%)	10 (45.45%)	12 (54.54%)	0 (0%)	-	
Cell Preservation	Lavage	8 (34.78%)	6 (26.08%)	4 (17.39%)	5 (21.74%)	-	5 No cells
	Swabbing	0 (0%)	4 (18.18%)	12 (54.54%)	6 (27.27%)	-	
Epithelial Cells	Lavage	6 (21.42%)	10 (35.71%)	4 (14.29%)	4 (14.29%)	4 (14.29%)	
	Swabbing	0 (0%)	12 (54.54%)	6 (27.27%)	4 (18.18%)	0 (0%)	
Individual Epithelial Cells	Lavage	11 (39.29%)	8 (28.57%)	1 (3.57%)	5 (17.86%)	3 (10.71%)	
	Swabbing	0 (0%)	10 (45.45%)	10 (45.45%)	2 (9.09%)	0 (0%)	
Clusters Epithelial Cells	Lavage	7 (25%)	6 (21.43%)	7 (25%)	2 (7.14%)	6 (21.42%)	
	Swabbing	0 (0%)	2 (9.09%)	8 (36.36%)	10 (45.45%)	2 (9.09%)	

**Table 6 animals-11-00088-t006:** Grading and distribution of the cytological characteristics of the cells in samples according to uterine swabbing or the uterine lavage method. No statistically significant differences were observed for any of these variables.

Cell Characteristics	Technique	Score 0	Score 1	Score 2	Score 3	Score 4	Comments
Acinar Arrangement	Lavage	18 (81.81%)	1 (4.55%)	3 (13.64%)	0 (0%)	-	5 No cells/1 not evaluable
	Swabbing	16 (72.72%)	4 (18.18%)	2 (9.09%)	0 (0%)	-	
Shape	Lavage	15 (68.18%)	3 (13.64%)	4 (18.18%)	-	-	5 No cells/1 not evaluable
	Swabbing	22 (100%)	0 (0%)	0 (0%)	-	-	
Pyknotic	Lavage	13 (59.09%)	9 (40.90%)	0 (0%)	0 (0%)	0 (0%)	5 No cells/1 not evaluable
	Swabbing	18 (81.81%)	4 (18.18%)	0 (0%)	0 (0%)	0 (0%)	
Cilia	Lavage	21 (95.45%)	1 (4.54%)	0 (0%)	0 (0%)	-	5 No cells/1 not evaluable
	Swabbing	22 (100%)	0 (0%)	0 (0%)	0 (0%)	-	
Vacuoles	Lavage	16 (72.72%)	3 (13.64%)	3 (13.64%)	0 (0%)	-	5 No cells/1 not evaluable
	Swabbing	22 (100%)	0 (0%)	0 (0%)	0 (0%)	-	
Nucleoli	Lavage	17 (77.27%)	2 (9.09%)	3 (13.64%)	0 (0%)	-	5 No cells/1 not evaluable
	Swabbing	20 (90.90%)	1 (9.09%)	1 (9.09%)	0 (0%)	-	
Mitosis	Lavage	21 (95.45%)	1 (4.54%)	0 (0%)	0 (0%)	-	5 No cells/1 not evaluable
	Swabbing	22 (100%)	0 (0%)	0 (0%)	0 (0%)	-	

**Table 7 animals-11-00088-t007:** Total numbers of cases and percentages per category for polymorphonuclear neutrophils (PMN) using each method. Category 0 corresponds to the absence of polymorphonuclear neutrophils (PMN) in 10 fields at 40× magnification. Category 1+ corresponds to a total count of 1 to 10 PMNs. Category 2 corresponds to a PMN count >10, which are individually dispersed. Category 3+ corresponds to a PMN count >10, which are distributed in clumps.

	Category 0	Category 1+	Category 2+	Category 3+
Swabbing	2 (9%)	8 (36%)	11 (50%)	1 (5%)
Lavage	9 (32%)	11 (39%)	8 (29%)	0 (0%)

## Data Availability

The data presented in this study are available on request from the corresponding author.

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
