# Peer review of "Endometrial Status in Queens Evaluated by Histopathology Findings and Two Cytological Techniques: Low-Volume Uterine Lavage and Uterine Swabbing"

_animals, 2021, doi:10.3390/ani11010088_

Round 1
Reviewer 1 Report
This study could be interesting, but I cannot understand the usefulness of the combination of endometrial cytology and uterine culture to evaluate the endometrial status of queens if these two methods cannot be applied in vivo since the small size of the uterus does not allow to obtain a representative diagnostic sample. The histological evaluation of the uterus removed after ovariohysterectomy remains the technique of choice to evaluate the possible presence of pathological processes such as endometritis. Therefore I suggest that you review the title and purpose of the work.
Author Response
This study could be interesting, but I cannot understand the usefulness of the combination of endometrial cytology and uterine culture to evaluate the endometrial status of queens if these two methods cannot be applied in vivo since the small size of the uterus does not allow to obtain a representative diagnostic sample. The histological evaluation of the uterus removed after ovariohysterectomy remains the technique of choice to evaluate the possible presence of pathological processes such as endometritis. Therefore I suggest that you review the title and purpose of the work.
Answer: Thank you very much for your comments and suggestions. You are partially right when stating that some of these techniques are not be clinically applied in vivo in queens. Endometrial swabbing is for sure an impossible technique to be performed in queens. However, uterine lavage is not so impossible to perform under a proper sedation status and endoscopy. In fact, a study published by Zambelly et al. in 2015 (First deliveries after estrus induction using deslorelin and endoscopic transcervical insemination in the queen. Theriogenology, 84, 773-778) brings to light the possibility of obtaining a uterine lavage sample by means of an endoscope. Maybe our mistake was not to expose in a clearer way that our objective was to establish which one the used techniques, either swabbing or lavage, was more representative of the current status of the endometrium.
On the other hand, regarding to your concern about uterine culture, this is a very common diagnosis technique in cases of equine endometritis and the authors considered important to include it in the study as a “control diagnosis technique”.
Reviewer 2 Report
General comments:
- Microbiological examination can confirm/exclude infection not inflammation. Please remember, endometritis is an inflammatory process, and infiltration of the endometrium by leucocytes detected by histhopathology or cytology confirms this diseases. Positive results of microbiological examination can confirm the etiology of inflammation but not inflammation/endometritis. Please remember about it during revision.
- Please provide more information in the ,,Introduction'' about the importance of endometritis in feline reproduction. Based on results presented in this study endometritis is not common in cats.
- Hypothesis is missing.
- I think that collection of cellular material from the uterus using swab technique is not possible in patients. However, I can imagine that low volume lavage is possible. What is your opinion about it? Do you see the possibility of using this research in practice? This problem should be mentioned in the ,,Discusion’’.
Specific comments:
- Title: This work is also focused on histological evaluation of the endometrium, and this aspect should be mentioned in the title.
- ,,Simple summary'' is missing.
- Line 23: UB?
- Lines 29 and 32: ,,US'' abbreviation is not explained.
- Lines 27-28. See general comments, point 1.
- Lines 52-55. See comments above. Positive results of microbiological examination is not equal endometritis.
- Line 57. Change ,,biopsy'' into ,,histopathological examination,,.
- Lines 56-59. These sentences are difficult to understand.
- Line 65. Consider to change: ,,cytological techniques'' into ,,method of sample collection,,.
- Line 79. Should be ,,kg''.
- Line 119.Please provide information about the intra and inter-assay values.
- Lines 149-150. Did you count PMN cells in stratum spongiosum and the stratum compactum?
- Line 175. Explain ,,40x ,,
- Lines 206-214. How did you evaluate inflammation? Only by cytology? It would be perfect to use results of histopathological examination as a gold standard, and next calculate sensitivity and specificity for results of cytological examinations.
- Table 3. Please provide the number of animals ovulated and non-ovulated.
- Lines 244-245. Please remove this sentence: Animals with higher values of serum progesterone also showed higher epithelium values (data not shown).
- Table 4. Description of this table is not clear. Group of animals with high and low progesterone concentrations should be indicated. P values; change commas into dots.
- Caption of Figure 4. Please remove the last sentence.
- Table 5 and 6. Please remove from the description ,, (P< 0.05)’’, and change commas into dots.
- Caption of Figure 5. Please remove (A) and (B), and P value is missing.
- Lines 261-266. Please rewrite. I propose for example: The number of PMN cells was no statistically significant between evaluated methods (Table 7).
- Lines 299-302: ???
- Please insert tables and figures in appropriate place in the text.
- Line 340. Infiltration of endometrium by PMN cells is classified as acute endometritis not subacute.
- Lines 340-344. ,,Inflammation’’ should be replaced by ,,infiltration’’ (3 times)
- Line 372. The comparison between queens and mares is questionable.
- Line 381, is it ok: [39, 40, 1]. One at the end?
Author Response
- Microbiological examination can confirm/exclude infection not inflammation. Please remember, endometritis is an inflammatory process, and infiltration of the endometrium by leucocytes detected by histhopathology or cytology confirms this diseases. Positive results of microbiological examination can confirm the etiology of inflammation but not inflammation/endometritis. Please remember about it during revision.
Answer: Thanks for your comment. You are right when stating that endometritis is defined as an inflammatory process and, in fact, the actual role of bacteria is not widely described in queens to date. However, a manuscript from Lawer et al., (Lawler D.F., Evans R.H., Reimers T.J., Colby E.D., Monti K.L. Histopathologic features, environmental factors, and serum estrogen, progesterone, and prolactin values associated with ovarian phase and inflammatory uterine disease in cats. Am J Vet Res. 1991;52(10):1747–1753) reported inflammatory uterine disease or infertility in 44 queens, being Escherichia coli the most isolated microorganism from the uterus of these females. This report suggests that maybe microorganisms may play a role in inducing endometritis in the queen. On the other hand, microorganisms-induced endometritis has also been described in different extent in other species such bitches, mares and cows.
- Please provide more information in the ,,Introduction'' about the importance of endometritis in feline reproduction. Based on results presented in this study endometritis is not common in cats.
Answer: The introduction have been modified following your suggestion (L50-52).
- Hypothesis is missing.
Answer: We presented the objectives of the study in substitution of the hypothesis
- I think that collection of cellular material from the uterus using swab technique is not possible in patients. However, I can imagine that low volume lavage is possible. What is your opinion about it? Do you see the possibility of using this research in practice? This problem should be mentioned in the ,,Discusion’’.
Answer: Thanks for your comment. As you have stated, swabbing technique is not possible to perform in queens, while low lavage volume is. In fact, previous studies (Zambelli D., Bini C., Cunto M. Endoscopic transcervical catheterization in the domestic cat. Reprod Domest Anim. 2015;50:13–16.) have already demonstrated the feasibility of transcervical cannulation. Following your advice, it has been mentioned in the discussion chapter (L408-411).
Specific comments:
- Title: This work is also focused on histological evaluation of the endometrium, and this aspect should be mentioned in the title.
Answer: Histology is included in the title
- ,,Simple summary'' is missing.
Answer: It has been added later in the submission
- Line 23: UB?
Answer: It is a mistake. The correct option should be US and has been already corrected in the text
- Lines 29 and 32: ,,US'' abbreviation is not explained.
Answer: Thanks for your comment. It was a mistake linked with your previous comment. US means uterine swabbing
- Lines 27-28. See general comments, point 1.
Answer: An answer has been provided in point 1 of the general comments
- Lines 52-55. See comments above. Positive results of microbiological examination is not equal endometritis.
Answer: Thanks for your comment. The role of microorganisms in endometritis has been previously described in other species such as bitches, mares and cows. And although scarce, some studies suggest a possible role of microorganisms in queen endometritis.
- Line 57. Change ,,biopsy'' into ,,histopathological examination,,.
Answer: Thanks for your suggestion. The change has been made
- Lines 56-59. These sentences are difficult to understand.
Answer: Slight modifications have been done in order to simplify the idea.
- Line 65. Consider to change: ,,cytological techniques'' into ,,method of sample collection,,.
Answer: Since cytology is the main aspect of this study, we have modified as follows: cytological sampling method.
- Line 79. Should be ,,kg''.
Answer: It has been corrected.
- Line 119.Please provide information about the intra and inter-assay values.
Answer: Information has been added.
- Lines 149-150. Did you count PMN cells in stratum spongiosum and the stratum compactum?
Answer: Yes, both stratum compactum and spongiosum were included in the counting.
- Line 175. Explain ,,40x ,,
Answer: It has been explained in the text.
- Lines 206-214. How did you evaluate inflammation? Only by cytology? It would be perfect to use results of histopathological examination as a gold standard, and next calculate sensitivity and specificity for results of cytological examinations.
Answer: In fact, microbiologiocal analyses and histopathological findings were used as controls to confirm the absence of infection/inflammation, ruling out thus the presence of endometritis.
- Table 3. Please provide the number of animals ovulated and non-ovulated.
Answer: This information has been added.
- Lines 244-245. Please remove this sentence: Animals with higher values of serum progesterone also showed higher epithelium values (data not shown).
Answer: The sentences has been deleted.
- Table 4. Description of this table is not clear. Group of animals with high and low progesterone concentrations should be indicated. P values; change commas into dots.
Answer: Thank you very much for detecting such a mistake. We had problems with the table format due to fact that authors work with different softwares (windows and mac). So, probably, all the “come and back” with the files made us not be aware of this big mistake. It has been fixed now.
- Caption of Figure 4. Please remove the last sentence.
Answer: The sentence has been removed.
- Table 5 and 6. Please remove from the description ,, (P< 0.05)’’, and change commas into dots.
Answer: (P<0.05) has been removed and commas have been changed into dots.
- Caption of Figure 5. Please remove (A) and (B), and P value is missing.
Answer: A and B letters have been removed, whereas P values has been added.
- Lines 261-266. Please rewrite. I propose for example: The number of PMN cells was no statistically significant between evaluated methods (Table 7).
Answer: The sentence has been re-written.
- Lines 299-302: ???
Answer: This has been corrected. Apparently, during the copy-paste process to the template, half of the image legend was displaced. In addition, a wrong Figure 3 had been added. It has removed and substituted by the correct one.
- Please insert tables and figures in appropriate place in the text.
Answer: Thanks for your recommendation. However, since we are not sure about the editing process of the journal, we have not modified it yet. In fact, most of the journals perform they specific editing process. Anyway, if it is something that has be done by the authors, please do not hesitate to let us know.
- Line 340. Infiltration of endometrium by PMN cells is classified as acute endometritis not subacute.
Answer: It has been changed following your advice.
- Lines 340-344. ,,Inflammation’’ should be replaced by ,,infiltration’’ (3 times)
Answer: Inflammation has been changed by infiltration
- Line 372. The comparison between queens and mares is questionable.
Answer: I totally agree with you. However, since literature in queens is so scarce, the authors opted to use some other species for comparing.
- Line 381, is it ok: [39, 40, 1]. One at the end?
Answer: It has been corrected
Round 2
Reviewer 2 Report
Accept in present form